# Ketogenic Diet Affects Sleep Architecture in C57BL/6J Wild Type and Fragile X Mice

**DOI:** 10.3390/ijms241914460

**Published:** 2023-09-22

**Authors:** Pamela R. Westmark, Aaron K. Gholston, Timothy J. Swietlik, Rama K. Maganti, Cara J. Westmark

**Affiliations:** 1Department of Neurology, University of Wisconsin, Madison, WI 53706, USA; prwestmark@wisc.edu (P.R.W.); agholston@wisc.edu (A.K.G.); tswietlik@wisc.edu (T.J.S.); maganti@neurology.wisc.edu (R.K.M.); 2Molecular Environmental Toxicology Center, University of Wisconsin, Madison, WI 53706, USA

**Keywords:** *Fmr1^KO^*, fragile X syndrome, ketogenic diet, electroencephalography (EEG), sleep

## Abstract

Nearly half of children with fragile X syndrome experience sleep problems including trouble falling asleep and frequent nighttime awakenings. The goals here were to assess sleep–wake cycles in mice in response to *Fmr1* genotype and a dietary intervention that reduces hyperactivity. Electroencephalography (EEG) results were compared with published rest–activity patterns to determine if actigraphy is a viable surrogate for sleep EEG. Specifically, sleep–wake patterns in adult wild type and *Fmr1^KO^* littermate mice were recorded after EEG electrode implantation and the recordings manually scored for vigilance states. The data indicated that *Fmr1^KO^* mice exhibited sleep–wake patterns similar to wild type littermates when maintained on a control purified ingredient diet. Treatment with a high-fat, low-carbohydrate ketogenic diet increased the percentage of non-rapid eye movement (NREM) sleep in both wild type and *Fmr1^KO^* mice during the dark cycle, which corresponded to decreased activity levels. Treatment with a ketogenic diet flattened diurnal sleep periodicity in both wild type and *Fmr1^KO^* mice. Differences in several sleep microstructure outcomes (number and length of sleep and wake bouts) supported the altered sleep states in response to a ketogenic diet and were correlated with altered rest–activity cycles. While actigraphy may be a less expensive, reduced labor surrogate for sleep EEG during the dark cycle, daytime resting in mice did not correlate with EEG sleep states.

## 1. Introduction

Fragile X syndrome (FXS) is a rare developmental disability with an estimated prevalence of 1 in 5000 [1]. Loss of expression of the RNA binding protein fragile X messenger ribonucleoprotein (FMRP) in FXS results in intellectual disability, autism, and seizures [1]. Many FXS phenotypes are manifested in *Fmr1^KO^* mice which lack expression of FMRP [2]; however, there has been difficulty translating promising therapeutics from the mouse model to human clinical trials. The development of validated outcome measures that translate between preclinical and clinical studies is a critical need for the FXS community [3]. Here we examined sleep architecture in *Fmr1^KO^* mice comparing EEG sleep measures to published actigraphy findings. We also explored the effects of ketogenic diet (KD) therapy, which has been shown to reduce hyperactivity and seizures in *Fmr1^KO^* mice [4], on sleep measures. We hypothesize that sleep states will be altered in *Fmr1^KO^* mice and rescued with KD.

Parents report that nearly half of children with FXS experience sleep difficulties with the most frequent issues involving sleep onset insomnia, reduced total sleep time and frequent awakenings [5,6,7,8,9,10,11,12]. Sleep disorders observed include obstructive sleep apnea (OSA) and nocturnal enuresis [6,12,13,14,15]. Polysomnography studies in FXS subjects showed a higher percentage of stage 1 non-rapid eye movement (NREM) sleep, a lower percentage of rapid eye movement (REM) sleep, and an increase in REM latency compared to normal controls indicative of disrupted sleep microstructure [16,17]. Considering the essential role that sleep plays in learning and memory consolidation, rescuing sleep deficits in FXS may be therapeutic, and sleep electroencephalography (EEG) and actigraphy may be viable outcome measures that translate between preclinical and clinical FXS studies. Thus, it is important to discern if there are altered sleep phenotypes in FXS preclinical models such as the *Fmr1^KO^* mouse.

The classic KD was introduced in 1921 to replace starvation in the treatment of seizures and is in use today to treat drug refractory epilepsy [18]. KDs are high in fat with moderate levels of protein and low carbohydrate. They force the body into a state of ketosis where ketones are used as the energy source instead of glucose. In recent years, the KD has become the new fad diet for weight loss and is being tested in numerous neurological models [19]. Recent studies in human and rodent models of autism indicate that the KD improves core behavioral symptoms [20,21,22,23,24,25,26,27,28]. There is an established link between circadian dysfunction and FXS-related proteins and accumulating evidence that the KD rewires the circadian clock [29,30,31,32,33,34,35,36,37,38,39]. Circadian rhythms are the 24-h cycles generated by a master pacemaker located in the suprachiasmatic nuclei (SCN) of the anterior hypothalamus of mammals. They control locomotor activity, feeding behavior, sleep–wake patterns, and other physiological and metabolic pathways. The importance of diet and circadian rhythm to sleep behavior are well recognized [30,35]. The KD decreases the hyperexcitability of in vitro epileptic networks, improves diurnal rhythmicity in epileptic mice, and improves slow-wave sleep and sleep quality in children with refractory epilepsy [39,40,41,42]. These findings prompt the hypothesis that the KD may be beneficial in attenuating sleep disturbances in FXS.

This study compares sleep states acquired by sleep EEG to published rest–activity cycles attained by actigraphy. EEG is the gold standard clinical tool employed in sleep research [43]; however, the clinical feasibility of overnight polysomnography is reduced in persons with developmental disabilities. While actigraphy is a convenient and indirect way of assessing sleep measures, details on sleep microstructure cannot be obtained. Actigraphy offers several advantages over polysomnography for preclinical research including 24-7 continuous recordings, ease of use, less invasive, and reduced cost [44]. Sleep measured by EEG can provide details of the sleep microarchitecture.

## 2. Results

### 2.1. Study Design

WT and *Fmr1^KO^* male littermate mice were randomly weaned onto control and KD at postnatal day 18 (P18), maintained on respective diets throughout the study, and tested for EEG sleep phenotypes at 5–6 months of age (Figure 1 and Appendix A). Similar to our prior study [4], there was significantly reduced body weight and increased blood ketones in WT and *Fmr1^KO^* mice in response to KD (Appendix A). EEG recordings were collected for 7 days and days 4 and 6 scored manually for sleep states (Appendix A, hypnograms).

### 2.2. Differences in 24-h Vigilance State Distribution Are Diet Dependent

Neither 24- nor 12-h time bins indicated WT versus *Fmr1* genotype-specific effects in the percent of time spent awake, asleep or in NREM or REM sleep in mice fed AIN-76A purified ingredient diet (Figure 2 and Appendix A (statistics)). Conversely, KD increased % NREM sleep by 13% in WT versus *Fmr1^KO^* mice. KD significantly reduced % time awake in both genotypes (21%↓ WT and 12%↓ *Fmr1^KO^*) and elevated % NREM (29%↑ WT and 18%↑ *Fmr1^KO^*) compared to AIN-76 controls.

### 2.3. KD Increases Time Spent in Sleep during the Dark Cycle

Mice are nocturnal with higher activity levels during the dark cycle and increased sleep during the light cycle. Binning the data by light and dark cycles indicated that KD increased time spent in NREM sleep during lights off in both genotypes, but significantly more in WT compared to *Fmr1^KO^*. Thus, a genotype-specific effect was observed when mice were fed KD with a 15% decrease in NREM sleep during the dark cycle in *Fmr1^KO^* compared to WT. The KD significantly decreased % wake time by 34% in WT and 23% in *Fmr1^KO^* and increased % NREM sleep by 79% in WT and 54% in *Fmr1^KO^* during the dark cycle compared to respective controls maintained on AIN-76A (Figure 2 and Appendix A (pie charts) and Appendix A). The KD decreased REM sleep by 27% in WT and 24% in *Fmr1^KO^* during the light cycle and increased REM sleep during the dark cycle by 84% in WT and 55% in *Fmr1^KO^*.

### 2.4. Sleep EEG Selectively Correlates with Actigraphy Findings during the Dark Cycle

We previously assessed rest–activity patterns in adult *Fmr1^KO^* mice by actigraphy in response to KD under the same treatment protocol used here (commencing at P18) and binning the data by 6-h increments (Figure 2D in [4]) and did not observe genotype-specific differences in rest–activity patterns. There were significantly decreased activity levels in both halves of the dark cycle (Zeitgeber hours, Z12–18 and Z18–24) and the first half of the light cycle (Z0–6) in both WT and *Fmr1^KO^* mice fed KD versus control diet. Activity levels were low in all cohorts during the second half of the light cycle (Table 1). Binning the EEG data into 6-h increments for direct comparison with the actigraphy data indicates concurrence regarding the absence of genotype-specific effects and decreased wake state during the dark cycle in WT and *Fmr1^KO^* mice fed KD (Appendix A and Appendix A). A point of convergence between the EEG and actigraphy data is that percent wake time during the first half of the light cycle did not correlate with activity levels as all cohorts exhibited equivalent % wake time by EEG but KD cohorts had decreased activity counts compared to AIN-76A cohorts.

Further division of the data into 2-h bins (Figure 3 and Appendix A) confirms decreased wake time in both WT and *Fmr1^KO^* mice fed KD during the dark cycle, specifically during the first half of the dark cycle and during the 2 h immediately before the transition to the light phase. There is also a genotype-specific difference with 26% increased wake time in *Fmr1^KO^* compared to WT mice fed KD during the final 2 h of the dark cycle. The KD blunts both wake state peaks during the dark period through increased NREM and REM sleep. The % NREM and % total sleep were the inverse of the wake data with highly statistically significant differences between all AIN-76A and KD cohorts during the dark cycle. Of note, there is a trend throughout the dark cycle that KD makes WT mice sleepier than *Fmr1^KO^* as evidenced by increased % NREM that reaches statistical significance during the final 2-h bin. REM sleep was significantly decreased during both halves of the light cycle and increased during both halves of the dark cycle in response to KD in both WT and *Fmr1^KO^* mice. Specifically, the KD blunted the REM peak (Z4–6 h, lights on) and the REM trough (Z16–18, lights off) in both WT and *Fmr1^KO^* mice. Overall, treatment with KD flattened diurnal sleep periodicity curves in both wild type and *Fmr1^KO^* mice, actigraphy results only concurred with sleep EEG during the dark cycle when mice should be awake, and EEG identified decreased REM sleep in response to KD during the light cycle irrespective of activity levels. There was a single 2-h bin (final 2 h of light cycle) where WT mice fed AIN-76A exhibited increased total % sleep compared to *Fmr1^KO^* fed AIN-76A.

### 2.5. KD Increases the Number of Wake, NREM and REM Bouts in WT Mice during the Dark Cycle

Sleep microarchitecture analysis included the number and length of wake, sleep, NREM and REM bouts over 24-h and 12-h bins. There were no genotype-specific differences in bout number in mice fed AIN-76A (Figure 4 and Appendix A). In mice treated with KD, *Fmr1^KO^* mice exhibited a decreased number of wake, sleep and NREM bouts over the 24-h period and during the dark cycle compared to WT. Diet-specific effects included statistically significant increases in wake, sleep and NREM bout number as a function of KD in WT mice over the 24-h period and during the 12-h dark cycle as well as an increased number of REM bouts in both WT and *Fmr1^KO^* during the dark cycle. In addition, sleep bout number was increased in WT mice with KD during the light cycle and REM bout number was decreased in *Fmr1^KO^* mice with KD during the light cycle.

### 2.6. KD Decreases the Length of Wake Bouts in Mice during the Dark Cycle

The average length of wake bouts during the 12-h dark cycle was significantly reduced in response to KD in WT and *Fmr1^KO^* mice (Figure 5 and Appendix A). The average reduction was 157 s for WT and 93 s for *Fmr1^KO^*. The reduced length of wake bouts in conjunction with an increased or equivalent number of wake bouts during the dark cycle indicates that the length of the wake bouts underlies reduced % wake time in response to KD during the dark phase. The average length of sleep bouts during the light cycle was significantly lower in WT mice in response to KD, which in conjunction with an increased number of sleep bouts, equates with equal % sleep in WT mice irrespective of diet. The average length of NREM bouts during the dark cycle was significantly higher in *Fmr1^KO^* mice in response to KD, which in conjunction with an equivalent number of bouts, indicates that increased % NREM sleep is primarily due to increased bout length whereas increased % NREM sleep during the dark cycle in WT mice in response to KD is primarily attributed to increased bout number.

Analysis of the single longest wake bout over the 24-h period indicates a 64% and 45% decrease in bout length in WT and *Fmr1^KO^* mice, respectively, in response to KD (Appendix A and Appendix A). There was also a 13% decrease in the longest REM bout in WT mice in response to KD over the 24-h period. KD caused a statistically significant average decrease of 19 min in the latency time to the first sleep bout in WT mice (Appendix A and Appendix A).

During the EEG scoring for sleep states, an EEG pattern was observed that suggested the mice were trying to enter REM but failing. To be scored as REM, the EEG pattern needed to exhibit seven or more epochs (an epoch is a 4-s period) of low amplitude/high frequency waves. Upon secondary analysis of the EEG recordings, instances of 1–6 epochs of low amplitude/high frequency waves were tallied as failed attempts to enter REM. The percentage of failed REM attempts indicates a statistically significant 20% decrease in failed REM attempts in *Fmr1^KO^* mice compared to WT fed AIN-76A during the dark cycle (Figure 6). This was the only genotype-specific EEG phenotype observed with the control diet.

### 2.7. Fmr1^KO^ Mice Do Not Exhibit Epileptic Activity on Sleep EEG

Epilepsy is a comorbid phenotype in FXS that manifests as benign focal epilepsy of childhood (BFEC) with predominantly nocturnal focal motor seizures and a characteristic EEG pattern with morphologically distinct centrotemporal discharges activated by sleep [45,46]. Similar to Pietropaolo and colleagues [47], we did not observe seizures, absences or interictal events in any WT or *Fmr1^KO^* mouse in scored EEG recordings.

## 3. Discussion

The identification of comparable outcome measures that can be feasibly employed in both preclinical mouse and clinical human studies is an important need for the FXS field [48]. Considering the pivotal role sleep plays in memory consolidation and its association with epileptiform activity, dysfunctional sleep in FXS may underly numerous disease phenotypes. We assessed sleep states in *Fmr1^KO^* mice by EEG and asked if actigraphy could serve as an easier implemented, less expensive surrogate for sleep EEG. We found similar sleep–wake patterns in wild type and *Fmr1^KO^* mice maintained on a control diet, and that sleep states did not correlate with activity levels during the light cycle when nocturnal animals sleep. These data suggest that sleep is not a viable preclinical outcome measure that translates between mouse and human models of FXS and that actigraphy, while a viable outcome measure, is not an absolute surrogate for sleep EEG in mice in our experience.

Mouse and fly models of FXS exhibit altered activity states and circadian rhythms, which potentially mirror the sleep disturbances observed in patients with the disorder. FMRP-related proteins are associated with increased [2,47,49,50,51,52,53,54,55,56,57,58,59,60,61,62,63,64,65,66,67,68,69,70,71,72,73], decreased [74], and no change [47,54,59,70,75,76,77,78,79,80,81,82,83,84] in measures of locomotor activity in *Fmr1^KO^* model organisms. Of note, testing in *Fmr1^KO^* mice was conducted by Saré and colleagues who employed a home-cage monitoring system that assessed activity counts as a surrogate for sleep during the light and dark phases as a function of age over six consecutive days where sleep was defined as 40 s of inactivity [85]. Juvenile *Fmr1^KO^* mice (P21) do not differ from controls. There is reduced “sleep” in adult *Fmr1^KO^* mice (P70 and P180) selectively during the light phase compared to controls. Thus, “sleep” disturbances emerge during the later stages of brain maturation. Their methodology resembles our actigraphy assessment of rest–activity in *Fmr1^KO^* mice where we observe hyperactivity in *Fmr1^KO^* mice [4], which concurs with most of the literature. Neither study found genotype-specific differences in habituation to the novel housing conditions. Both studies found increased activity in adult mice selectively during the light phase compared to controls, albeit we selectively found the difference during the first half of the light cycle and lost statistical significance if we binned in 12-h timeframes. Saré and colleagues found a 6–8% increase in activity counts during the 12-h light cycle in adult *Fmr1^KO^* mice, and we found a 38% increase during the first 6 h of the light cycle. Numerous environmental factors can affect the outcome and magnitude of rodent behavioral studies. The disadvantage of actigraphy testing as a surrogate for sleep is lack of data on sleep stages and bout duration, while the disadvantages of EEG testing are the invasiveness of the surgery, implantation of electrodes, weight and restriction of the wires attached to the headcap, and the inability to test juvenile mice due to limitations with attaching and maintaining the electrode headcap.

Recent studies in FXS human and rodent models identify enhanced resting state gamma power and reduced inter-trial coherence of sound-evoked gamma oscillations as common EEG biomarkers [86,87,88,89,90,91,92,93,94,95,96,97]. There is also evidence of exaggerated theta power in subjects with FXS and abnormal hippocampal theta–gamma phase amplitude coupling in *Fmr1^KO^* mice [98,99,100]. Furthermore, in vivo electrophysiological studies in the hippocampal CA1 region of *Fmr1^KO^* mice indicate a deficit in REM sleep due to a reduction in the frequency of REM bouts (60 min test session during the light phase), which is consistent with sleep architecture abnormalities in FXS patients [101]. Considering the documented prevalence of sleep problems in FXS, we were surprised we did not find differences in REM or NREM sleep in *Fmr1^KO^* mice over 24-h test periods. Our three-channel EEG methodology is less precise than the published 18-tetrode microdrive technology but also less invasive. We did not find altered wake or sleep (NREM, REM) states as a function of *Fmr1* genotype with electrodes implanted in the right frontal and left parietal cortices of the brain and with the mice maintained on a control AIN-76A diet. We also did not find differences in the number and/or length of wake, NREM or REM bouts as a function of *Fmr1* genotype. While scoring EEG, we observed a pattern whereby mice appeared to be trying to enter REM but failing. This was the only genotype-specific difference we found between WT and *Fmr1^KO^* mice maintained on AIN-76A diet from weaning.

An important question regarding EEG outcomes is response to treatment. Prior studies assessing sleep by home-cage monitoring found that the GABA_B_ agonist R-baclofen and the mTOR inhibitor rapamycin did not rescue deficits in the light phase [85,102]. In fact, rapamycin reduced “sleep” duration and had adverse effects on social behavior in both adult WT and *Fmr1^KO^* mice. Herein, we also find that total sleep, as assessed by EEG, did not differ during the light cycle with KD therapy, which is highly effective at reducing audiogenic-induced seizures (AGS) in *Fmr1^KO^* mice tested during the light cycle [4]. NREM sleep did not change with KD during the light cycle, but REM sleep was reduced. We chose to test the effect of KD on sleep EEG in *Fmr1^KO^* mice because the KD is proving therapeutic for many disease states that are comorbid with FXS, including epilepsy and autism; individuals with FXS present with an abnormal fatty acid profile [103]; and our prior work indicated that KD was as or more effective than the best metabotropic glutamate receptor 5 (mGluR_5_) inhibitors tested in the AGS assay.

We found that a KD increased NREM and REM sleep during the dark cycle in both WT and *Fmr1^KO^* mice, which corresponds to reduced activity levels in the actigraphy assay; however, REM sleep was reduced in mice in response to a KD during the light cycle, which did not correlate with activity counts. These data suggest that the mice were awake longer but not moving during the light cycle when nocturnal animals should be sleeping. BAER-101, a selective GABA_A_ agonist, reduces delta power in *Fmr1^KO^* mice during the light cycle during a 5-min period free of excessive movement [104]. Others have demonstrated states of quiet rest or quiet wakefulness in WT rats (3–9 Hz activity) when the *Fmr1^KO^* rat cortex remains activated (18–52 Hz) [105]. Thus, the KD and GABA agonists may increase quiet rest in mice. Decreased REM sleep during lights on is suggestive of keto insomnia, which is a lack of sleep reported by humans during the early days of KD introduction. Increased NREM and REM sleep in mice during lights off is suggestive of somnogenic (sleepiness) effects of the KD during a time when the mice are expected to be active. In contrast, in humans, a KD reduces nocturnal awakenings and improves excessive daytime sleepiness [106].

The mechanism underlying KD-induced changes in sleep architecture in C57BL/6J mice remain to be determined. Protein-enriched food (casein; three-fold over the standard 18% by weight), but not sugar or fat enrichment, has been shown to decrease the percent awakenings in response to stimulation with a corresponding decrease in the number of sleep and wake bouts [107]. Here, the KD, which has decreased protein (8.6%) versus the control diet (18%) caused increased percent awakenings (albeit without stimulation) as there was an increased number of wake, sleep, NREM and REM bouts during the dark cycle in WT C57BL/6J mice suggesting that dietary protein content affects arousal. Similarly, the length of sleep bouts and % REM were increased with high protein in the diet, and we found decreased length of sleep bouts and % REM during the light cycle with the KD, which contained lower protein. Thus, suboptimal protein content of the diet may affect multiple sleep macro- and microarchitecture phenotypes.

Alternatively, or in addition to protein, ketones may underly alterations in sleep structure in response to a KD. Others have tested the effects of a KD plus medium chain triglycerides (MCT) on sleep in mice and found increased ketone levels and reduced REM sleep duration during the light cycle [108]. Their treatment started at 8 weeks of age, and they did not observe decreased body weight. NREM sleep and wakefulness were not affected, but slow wave sleep (delta/theta ratio) during NREM increased during the active phase with KD-MCT. We also observed increased ketones and reduced REM sleep during the light cycle, but we have significantly reduced body weight. In addition, our % wake and % NREM sleep during the light cycle were also unaffected and our % NREM increased during the dark cycle. As the two studies have similar findings, the data suggest that our significantly reduced body weight in response to KD is not a confounding factor interfering with sleep architecture.

Aberrant signaling through metabotropic glutamate receptor 5 (mGluR_5_), the most studied therapeutic target for FXS [109], may contribute to sleep phenotypes in FXS. Knockout mice lacking mGluR_5_ do not exhibit altered sleep phenotypes during the light cycle under baseline conditions; however, they do exhibit lack of recovery sleep after sleep deprivation including decreased REM and NREM sleep compared to WT littermates [110]. Similarly, we did not find differences in REM or NREM sleep between WT and *Fmr1^KO^* mice fed control diet, but the KD diet caused a trend for decreased % NREM throughout the dark cycle in *Fmr1^KO^* versus WT that reached statistical significance around the dark/light transition (Z22–24 and Z2–4). Thus, stress whether in the form of sleep deprivation or a stringent diet, may exacerbate sleep deficits in *Fmr1^KO^* mice. *Fmr1^KO^* mice overexpress amyloid precursor protein (APP) through an mGluR_5_-dependent pathway [111], and the J20 mouse model of Alzheimer’s disease that over-expresses APP with familial mutations exhibits a deficit in REM sleep, which is rescued with the mGluR_5_ inhibitor CTEP. CTEP also improves NREM delta and sigma power but did not correct rest–activity rhythms or alter other behavioral outcomes [112]. The effect of mGluR_5_ inhibitors on sleep states in *Fmr1^KO^* mice remains to be determined.

The limitations of this study include testing adult animals, as larger deficits may be observed during postnatal development, as well as the labor-intensive requirement for manual scoring of the EEG recordings. We tested adult mice in this preliminary study due to technical limitations with the weight of the EEG headcaps compared to mouse body weight and because postnatal animals are rapidly growing, including the skull, which could interfere with headcap stability. We only tested male mice in this study due to the labor-intensive nature of manually scoring the sleep EEG, which required an average of 4–6 h per mouse per 24-h period scored, and because KD selectively attenuates seizures, reduces body weight, and decreases activity levels in male but not female *Fmr1^KO^* mice. We are in the process of scoring sleep EEG in adult *Fmr1^KO^* mice commencing KD therapy at 3 months of age, where *Fmr1^KO^* mice exhibit a 38% increase in hyperactivity during the first half of the light cycle (Figure 3 in [4]). Increased hyperactivity in *Fmr1^KO^* mice selectively during the beginning of the light cycle concurs with the sleep phenotype in the human disorder where children with FXS have trouble falling asleep. The lack of hyperactivity here in *Fmr1^KO^* mice treated from the time of weaning may be due to a therapeutic effect of the control diet (AIN-76A) compared to standard chows. The effects of diet on sleep architecture and resultant improvements in disease phenotypes remain to be determined in FXS models.

In conclusion, this study provides substantial data on sleep EEG outcomes in mice as a function of *Fmr1* genotype and KD. Several sleep outcome measures agreed with rest–activity data in *Fmr1^KO^* mice, but selectively during the dark cycle (mouse wake cycle) and not during the light cycle (mouse sleep time). These data suggest that actigraphy is not a viable surrogate to assess sleep states per se in mice. Actigraphy and sleep EEG outcomes during the mouse active cycle responded to KD treatment in the same direction. Thus, actigraphy may afford the ability to monitor mice of all ages over a longer period in preclinical studies without surgery, headcap implantation or risk of destabilization of the headcap, while wrist or ankle actigraphy bracelets could approximate resting versus active states and response to interventions in human clinical trials for FXS. The lack of evidence for improved sleep during the light cycle in *Fmr1^KO^* mice suggests that the ketogenic diet may not be beneficial in treating sleep problems associated with the disorder, albeit there are significant effects on seizures and activity levels.

## 4. Materials and Methods

### 4.1. Materials

A detailed list of materials with supplier information is provided (Appendix A).

### 4.2. Mice

The *Fmr1^tm4Cgr^* (*Fmr1^KO^*) mice were originally developed by the Dutch–Belgian FXS Consortium and backcrossed > 11 times to FVB mice [2]. They were backcrossed into the C57BL/6J background by Dr. Bill Greenough’s laboratory (University of Illinois at Urbana-Champaign) and distributed to other laboratories. We have maintained the *Fmr1^KO^* mice in the C57BL/6J background at the University of Wisconsin-Madison for over 15 years with occasional backcrossing with C57BL/6J mice from Jackson Laboratories to avoid genetic drift. Breeding pair for these experiments were housed in static microisolator cages with ad libitum access to food (Teklad 2019) and water in a temperature- and humidity-controlled vivarium on a 12-h light cycle. The bedding (Shepherd’s Cob + Plus, ¼ inch cob) contained nesting material as the only source of environmental enrichment. All animal husbandry, surgery and euthanasia procedures were performed under NIH and an approved University of Wisconsin-Madison animal care protocol administered through the Research Animal Resources Center with oversight from the Institutional Animal Care and Use Committee (IACUC). *Fmr1* genotypes were determined by PCR analysis of DNA extracted from tail biopsies with HotStarTaq polymerase (Qiagen Inc, Germantown, MD, USA; catalog #203205) and Jackson Laboratories’ (Bar Harbor, ME, USA) primer sequences oIMR2060 [mutant forward; 5′-CAC GAG ACT AGT GAG ACG TG-3′], oIMR6734 [wild type (WT) forward; 5′-TGT GAT AGA ATA TGC AGC ATG TGA-3′], and oIMR6735 [common reverse; 5′-CTT CTG GCA CCT CCA GCT T-3′], which produces PCR products of 400 base pairs (*Fmr1^KO^*) and 131 base pairs (WT). Heterozygote females exhibit both the 400 and 131 base pair bands. *Fmr1^HET^* females were bred with *Fmr1^KO^* males and generated WT and *Fmr1^KO^* male littermate mice for the described experiments. Experimental animals were derived from multiple breeding pairs as recommended [113,114]. Mice were randomly weaned onto control and KD at postnatal day 18 (P18), maintained on respective diets throughout the study, and tested for EEG sleep phenotypes at 5–6 months of age (Figure 1 and Appendix A). Some of the mice [AIN-76A (n = 3), WT fed a KD (n = 2), *Fmr1^KO^* fed AIN-76A (n = 6), and *Fmr1^KO^* fed a KD (n = 5)] completed a previously described KD treatment and behavioral battery (Figure 2 of [4]) before EEG. The remaining mice were generated to increase sample sizes and were experimentally naïve prior to EEG electrode implantation albeit diet treatments. Mice were socially housed prior to surgery and individually housed after EEG electrode implantation. Cohorts included AIN-76A (n = 12), WT fed KD (n = 9), *Fmr1^KO^* fed AIN-76A (n = 16), and *Fmr1^KO^* fed KD (n = 13).

### 4.3. EEG Electrode Implantation

Right frontal, left parietal, and occipital electrodes were placed along with two stainless steel wire electrodes in nuchal muscles for EMG under isoflurane anesthesia according to methods described previously [112] (Appendix A). The occipital electrode served as the ground and as the reference electrode for the surgical recording procedure. Mice were allowed to recover from surgery for 3 days prior to transfer to tethered EEG recording setups with mice housed individually in Plexiglas® chambers for sleep recording. EEG signals were acquired for 7 days (across the light and dark cycles) on an XLTEK machine (Natus, Madison, WI) with a 512 Hz sampling rate. EDF Browser software (v.2.40) was used to convert Natus EEG recordings into files compatible with Sirenia^®^ Sleep Pro software v.1.3.2 (Pinnacle Technology, Lawrence, KS, USA). EEG signals from days 4 and 6 were scored by hand with Sirenia^®^ Sleep Pro software v1.3.2 in 4-s epochs for wake, REM, and NREM states by two scorers blinded with respect to treatment group as previously described [112,115,116]. Waking epochs were identified as those with high EMG amplitude, and epochs with little EMG activity were scored as sleep. Specific sleep states were scored based on predominant EEG power where NREM was associated with low frequency and high amplitude delta (1–4 Hz) activity and REM was associated with increased frequency but low amplitude theta (5–7 Hz) activity. Sample EEG recordings scored for wake, NREM and REM states are shown in Appendix A with hypnograms for all scored mice provided in Appendix A. Data were binned into 2-, 6-, 12- and 24-h increments for analyses.

### 4.4. Biometrics

Ketone and glucose levels were assessed in urine or blood as indicated in the figure legend using a Precision Xtra blood glucose and ketone monitoring system (Abbott Diabetes Care Inc., Alameda, CA, USA). For urine samples, mice were gently restrained at the neck and urine collected. For blood samples, mice were anesthetized with isoflurane and blood collected from the inferior vena cava and assayed for ketone and glucose levels with a Precision Xtra meter. Low off-scale glucose meter readings were adjusted to 20 mg/dL. High off-scale glucose meter readings were adjusted to 500 mg/dL. High off-scale ketone meter readings were adjusted to 8.0 mmol/L. Blood samples were collected during the light phase after 4 h fasting.

### 4.5. Statistical Analysis

Data were analyzed with Microsoft 365 Excel^TM^ version 2306 and Prism version 10.0.0 (153) software. The vast majority of the data passed the D’Agostino & Pearson test for normal Gaussian distribution and all data were analyzed similarly with parametric ANOVA tests. Average data are presented ±SEM and statistical significance determined by 2-way ANOVA with Tukey’s multiple comparison test (Appendix A). The number of subjects, genotype and treatment details are included in the figure legends.

## Figures and Tables

**Figure 1 ijms-24-14460-f001:**
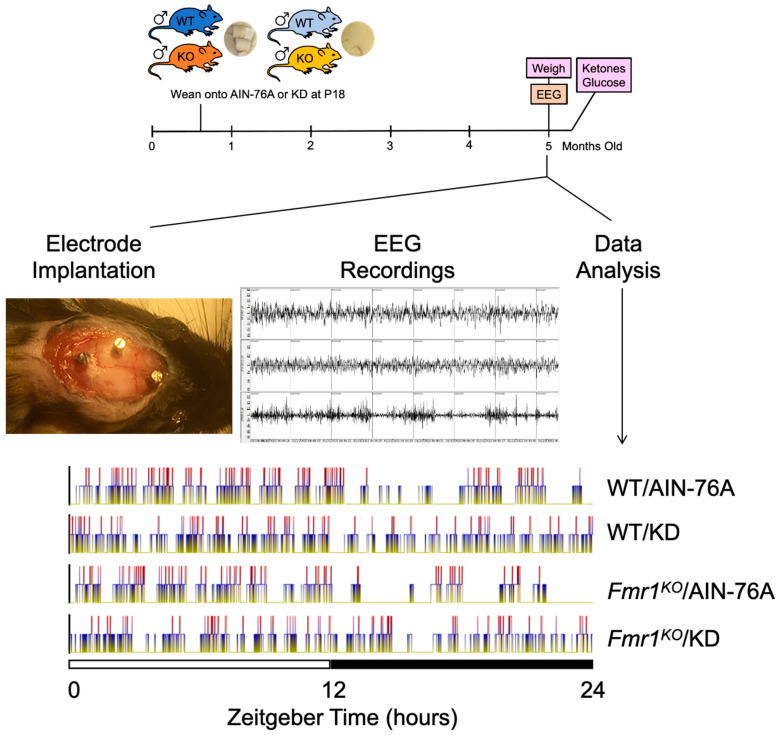
Study Design. Male WT and *Fmr1^KO^* mice were randomized to AIN-76A versus KD at P18. At 5 months of age, mice underwent surgery for EEG/EMG electrode implantation followed by EEG recordings over 7 days and data analysis. Mice were weighed on the day of surgery. Blood samples were tested for glucose and ketone levels at the time of euthanization. Representative sleep stage hypnograms as a function of genotype and diet are shown where yellow = wake state, blue = NREM sleep, and red = REM sleep.

**Figure 2 ijms-24-14460-f002:**
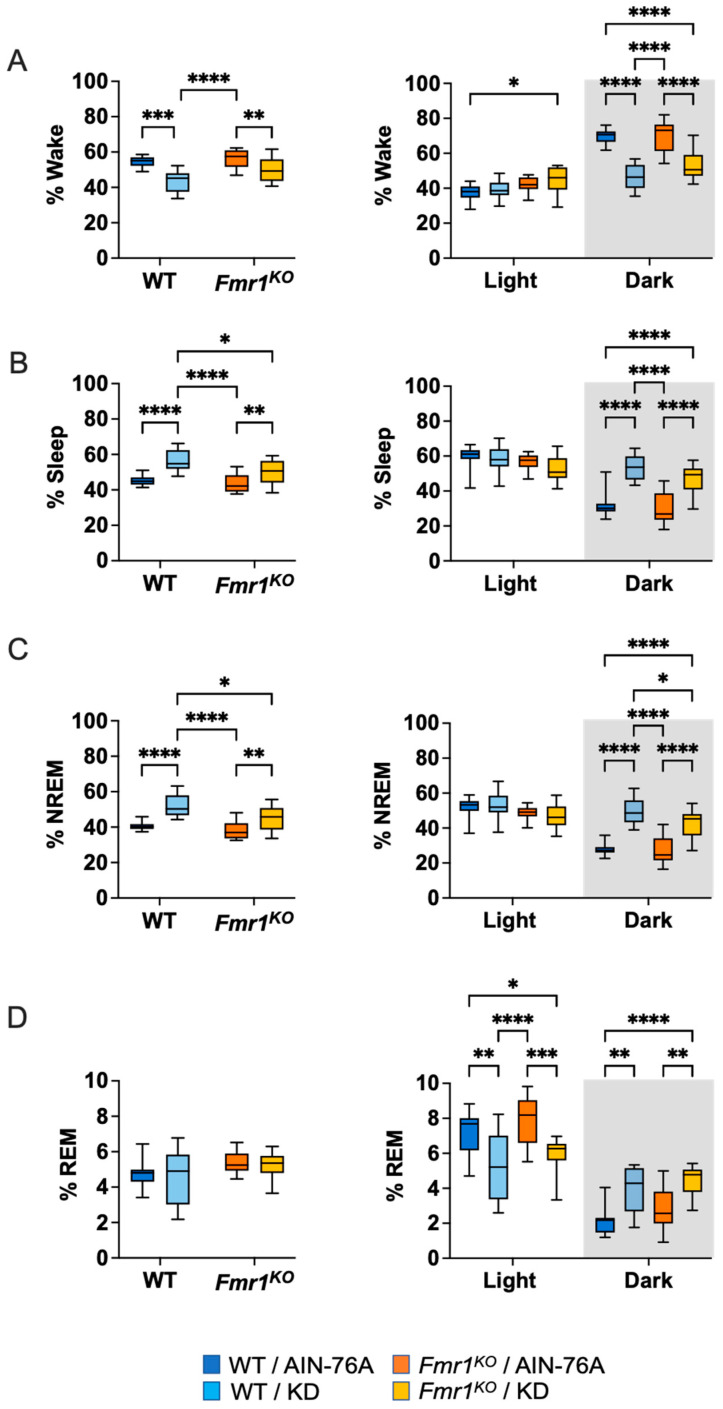
Effect of KD on Sleep Architecture in WT and *Fmr1^K^*^O^ Mice as a Function of Light and Dark Cycles. EEG/EMG recordings were scored for wake, sleep, NREM and REM activity and the data parsed into 24-h full day and 12-h light/dark bins starting at Zeitgeber time zero. The average percent sleep state was plotted versus bin for (**A**) % wake, (**B**) % sleep, (**C**) % NREM, and (**D**) % REM. Mouse cohorts included WT fed AIN-76A (n = 12), WT fed KD (n = 9), *Fmr1^KO^* fed AIN-76A (n = 16), and *Fmr1^KO^* fed KD (n = 13). Statistics with GraphPad Prism included 2-way ANOVA with post-hoc Tukey multiple comparison tests. Key for statistical significance: * *p* < 0.05, ** *p* < 0.01, *** *p* < 0.001, **** *p* < 0.0001.

**Figure 3 ijms-24-14460-f003:**
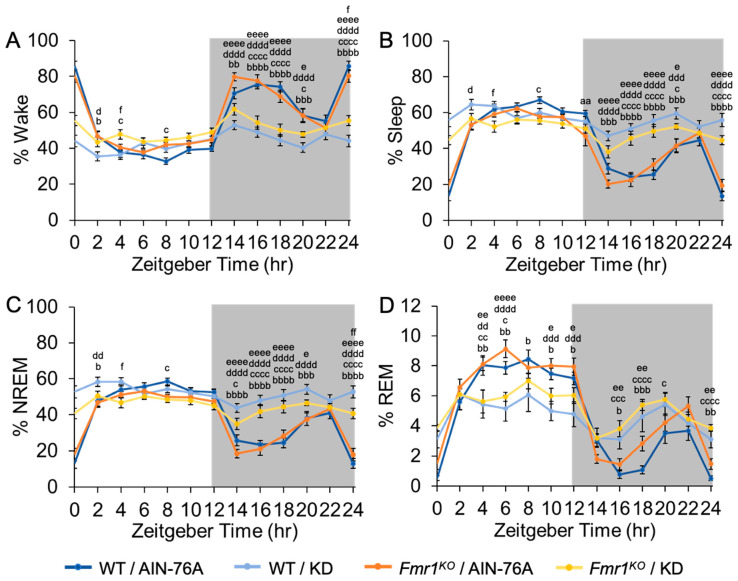
Effect of KD on Sleep Architecture in WT and *Fmr1^KO^* Mice as a Function of Time of Day. EEG/EMG recordings were scored for wake, sleep, NREM and REM activity and the data parsed into 2-h bins starting at Z0 (lights on). The average percent sleep state was plotted versus 2-h time bins for (**A**) % wake, (**B**) % sleep, (**C**) % NREM, and (**D**) % REM. Mouse cohorts included WT fed AIN-76A (n = 12), WT fed KD (n = 9), *Fmr1^KO^* fed AIN-76A (n = 16), and *Fmr1^KO^* fed KD (n = 13). Statistics with GraphPad Prism included 2-way ANOVA with repeated measures and post-hoc Tukey multiple comparison tests. Key for statistical significance: WT/AIN-76A versus *Fmr1^KO^*/AIN-76A = “a”, WT/AIN-76A versus WT/KD = “b”, WT/AIN-76A versus *Fmr1^KO^*/KD = “c”, *Fmr1^KO^*/AIN-76A versus WT/KD = “d”, *Fmr1^KO^*/AIN-76A versus *Fmr1^KO^*/KD = “e”, and WT/KD versus *Fmr1^KO^*/KD = “f”; and aa *p* < 0.01, b *p* < 0.05, bb *p* < 0.01, bbb *p* < 0.001, bbbb *p* < 0.0001, c *p* < 0.05, cc *p* < 0.01, ccc *p* < 0.001, cccc *p* < 0.0001, d *p* < 0.05, dd *p* < 0.01, ddd *p* < 0.001, dddd *p* < 0.0001, e *p* < 0.05, ee *p* < 0.01, eeee *p* < 0.0001, f *p* < 0.05, ff *p* < 0.01.

**Figure 4 ijms-24-14460-f004:**
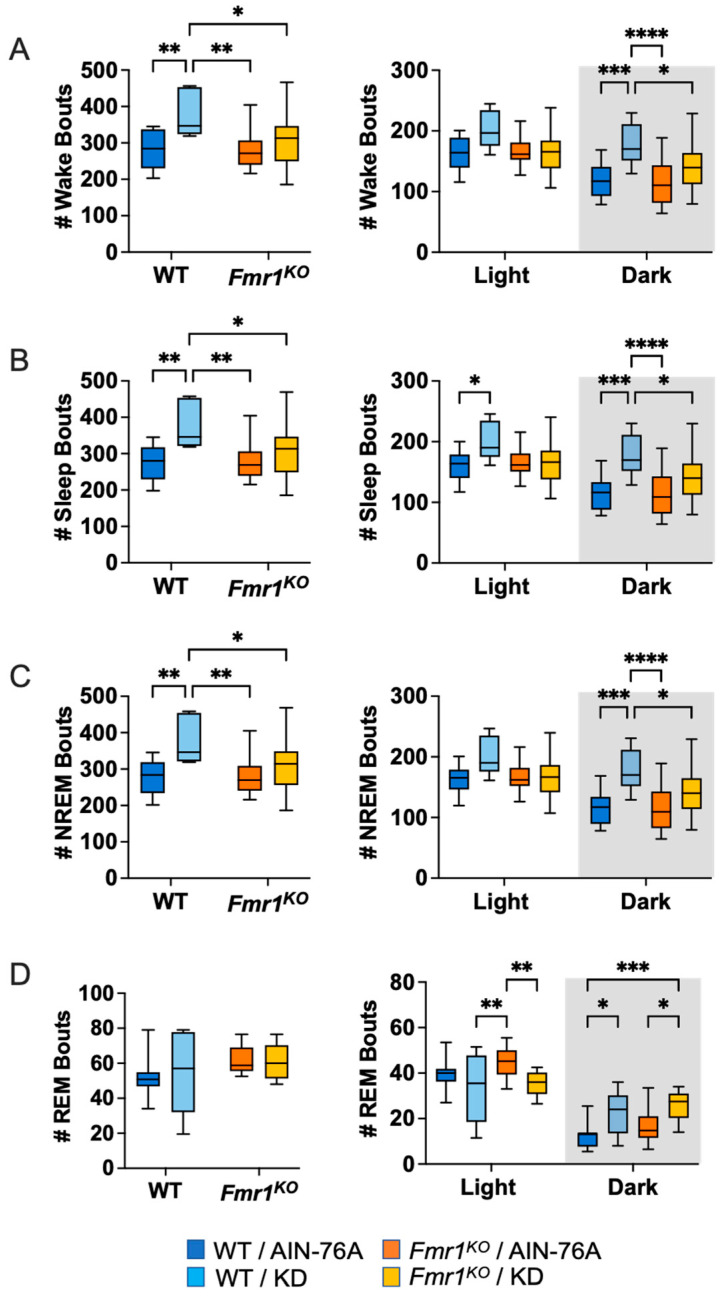
Effect of KD on Sleep Micro-Architecture (Number of Bouts) in WT and *Fmr1^K^*^O^ Mice as a Function of Light and Dark Cycles. EEG/EMG recordings were scored for wake, sleep, NREM and REM activity and the data parsed into 24-h full day and 12-h light/dark bins starting at Z0. The average number of sleep/wake state bouts was plotted versus bin for (**A**) wake bouts, (**B**) sleep bouts, (**C**) NREM bouts, and (**D**) REM bouts. Mouse cohorts included WT fed AIN-76A (n = 12), WT fed KD (n = 9), *Fmr1^KO^* fed AIN-76A (n = 16), and *Fmr1^KO^* fed KD (n = 13). Statistics with GraphPad Prism included 2-way ANOVA with post-hoc Tukey multiple comparison tests. Key for statistical significance: * *p* < 0.05, ** *p* < 0.01, *** *p* < 0.001, **** *p* < 0.0001.

**Figure 5 ijms-24-14460-f005:**
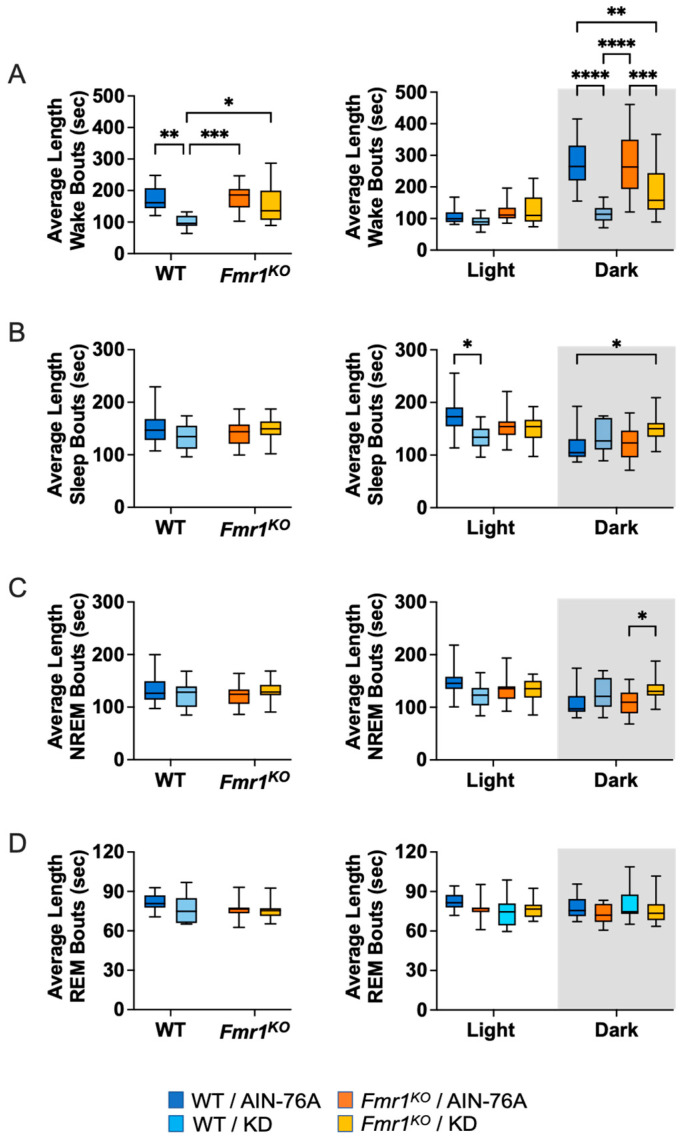
Effect of KD on Sleep Micro-Architecture (Length of Bouts) in WT and *Fmr1^K^*^O^ Mice as a Function of Light and Dark Cycles. EEG/EMG recordings were scored for wake, sleep, NREM and REM activity and the data parsed into 24-h full day and 12-h light/dark bins starting at Z0. The average length of time of sleep/wake bouts was plotted versus bin for (**A**) wake bouts, (**B**) sleep bouts, (**C**) NREM bouts, and (**D**) REM bouts. Mouse cohorts included WT fed AIN-76A (n = 12), WT fed KD (n = 9), *Fmr1^KO^* fed AIN-76A (n = 16), and *Fmr1^KO^* fed KD (n = 13). Statistics with GraphPad Prism included 2-way ANOVA with post-hoc Tukey multiple comparison tests. Key for statistical significance: * *p* < 0.05, ** *p* < 0.01, *** *p* < 0.001, **** *p* < 0.0001.

**Figure 6 ijms-24-14460-f006:**
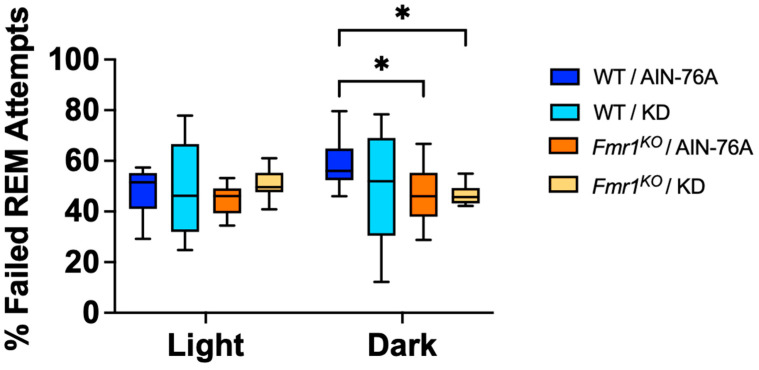
Effect of KD on Ability to Enter REM Sleep in WT and *Fmr1^K^*^O^ Mice. EEG/EMG recordings were scored for the number of failed attempts to enter REM and expressed as a percentage of total attempts. Mouse cohorts included WT fed AIN-76A (n = 12), WT fed KD (n = 9), *Fmr1^KO^* fed AIN-76A (n = 16), and *Fmr1^KO^* fed KD (n = 13). Statistics with GraphPad Prism included 2-way ANOVA with post-hoc Tukey multiple comparison tests. Key for statistical significance: * *p* < 0.05.

**Table 1 ijms-24-14460-t001:** Comparison of Rest/Activity with Sleep Phenotypes in Male WT and *Fmr1^KO^* Mice.

Treatment	Actigraphy [4]	Sleep EEG	Concur
KD start at P18			
Genotype effects	none (AIN-76A)	WT ↑ sleep (AIN-76A Z10–12)	-
	*Fmr1^KO^* ↑ activity (KD Z0–6) ([4] Figure 2D)	*Fmr1^KO^* ↑ wake and ↓ sleep, NREM (KD Z2–4) (Figure 3)	✓
		*Fmr1^KO^* ↑ wake and ↓ NREM (KD Z22–24) (Figure 3)	-
Diet effects	KD ↓ activity (WT and *Fmr1^KO^* Z0–6) ([4] Figure 2D)	KD ↓ REM Z0–12 (Figure 2 and Appendix A)	-
	KD ↓ activity (WT and *Fmr1^KO^* Z12–24) ([4] Figure 2D)	KD ↑ sleep, NREM, REM (WT and *Fmr1^KO^* Z12–24) (Figure 2 and Appendix A)	✓
KD start at 2–3 months of age			
Genotype effects	*Fmr1^KO^* ↑ activity (AIN76A Z0–6) ([4] Figure 3E)	TBD	-
Diet effects	KD ↓ activity (*Fmr1^KO^* Z12-24) ([4] Figure 3E)	TBD	-

## Data Availability

Data are contained within the article or Appendix A. Sirenia EEG/EMG files are available from the corresponding author upon request.

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
