# Peer review of "Ketogenic Diet Affects Sleep Architecture in C57BL/6J Wild Type and Fragile X Mice"

_ijms, 2023, doi:10.3390/ijms241914460_

Round 1

Reviewer 1 Report

In the current study, the author assessed sleep-wake cycles in mice in response to Fmr1 null genotype. Further, the therapeutic potential of the ketogenic diet (KD) was also studied in this genotype. Adult male mice were recorded for EEG during the light-dark cycle. Fmr1 KO mice exhibited sleep-wake patterns like wild-type littermates when maintained on a control purified ingredient diet. However, KD treatment disrupts the sleep-wake cycle in the dark cycle. The author further compared their results with previously reported actigraphy data sets. Overall study tested the novel hypotheses and is well designed.

A few concerns were reported as follows.

1)      As mentioned by the author that FXS patients exhibit sleep problems, however, no such differences were observed in Fmr1 KO mice over the 24-hour test period conducted in the current study. As indicated, the normal diet used in the current study may have a therapeutic effect compared to standard chows, therefore author should consider recording and comparing between standard chow vs normal diet.

2)      Study concluded that KO has adverse effects on dark cycle sleep, this raises the question of whether the KO diet is therapeutically relevant to rescue the sleep problem.

3)  As shown in Figure.1, 3 screws were used to tether the electrodes. 1 each for frontal and parietal cortices and the last occipital screw for ground. What is the reference electrode in the surgical recording procedure?

Author Response

Dear IJMS and Reviewers,

We would like to thank the reviewers and the editorial staff for the careful critique of our manuscript and the valuable suggestions for improvement. Please find below a copy of the reviewer comments followed by our responses.

Sincerely,

Cara Westmark

Reviewer 1

In the current study, the author assessed sleep-wake cycles in mice in response to Fmr1 null genotype. Further, the therapeutic potential of the ketogenic diet (KD) was also studied in this genotype. Adult male mice were recorded for EEG during the light-dark cycle. Fmr1 KO mice exhibited sleep-wake patterns like wild-type littermates when maintained on a control purified ingredient diet. However, KD treatment disrupts the sleep-wake cycle in the dark cycle. The author further compared their results with previously reported actigraphy data sets. Overall study tested the novel hypotheses and is well designed. Thank you for the positive review.

A few concerns were reported as follows.

1)      As mentioned by the author that FXS patients exhibit sleep problems, however, no such differences were observed in Fmr1 KO mice over the 24-hour test period conducted in the current study. As indicated, the normal diet used in the current study may have a therapeutic effect compared to standard chows, therefore author should consider recording and comparing between standard chow vs normal diet. Yes, we are planning to test the effects of standard diets. Due to the extensive time commitment for hand scoring EEG, additional experiments are beyond the scope of this paper.

2)      Study concluded that KO has adverse effects on dark cycle sleep, this raises the question of whether the KO diet is therapeutically relevant to rescue the sleep problem. We agree that this study raises questions regarding therapeutic efficacy of KD in FXS in regard to sleep problems. We have added the statement, “The lack of evidence for improved sleep during the light cycle in Fmr1KO mice suggests that the ketogenic diet may not be beneficial in treating sleep problems associated with the disorder, albeit there are significant effects on seizures and activity levels.” at the end of the Discussion.

3)  As shown in Figure.1, 3 screws were used to tether the electrodes. 1 each for frontal and parietal cortices and the last occipital screw for ground. What is the reference electrode in the surgical recording procedure? The occipital screw serves as both ground and reference. This has been clarified in the Methods.

Reviewer 2

The study is very interesting and deals with a very current topic.This study compares sleep states acquired by sleep EEG to published rest-activity cycles attained by actigraphy. The authors find similar sleep-wake patterns in wild type (WT) and Fmr1KO mice maintained on a purified ingredient, casein protein-based diet. KD disrupts sleep-wake patterns in both strains and reveals some genotype-specific differences. Although the manuscript is of interest to the scientific community, I have some suggestions for the authors. Thank you.

Introduction.

The introduction is a little confusing. The authors move from one topic to another in an unclear manner. For example, in the central part of this paragraph (lines 51-54), the authors seem to want to describe the purpose of the study; subsequently (lines 77-78) the authors report the purpose of the study again. I believe that authors should clearly specify the purpose of the study at the end of the introductory paragraph. Furthermore, in this section, it is not necessary to summarize the results as done in lines 78-81. We moved the purpose of the study to the end of the first paragraph of the Introduction and deleted lines 79-81.

I also think the authors could explain the ketogenic diet better; for example they can take inspiration from the following recent publications:

- Polito R., The Ketogenic Diet and Neuroinflammation: The Action of Beta-Hydroxybutyrate in a Microglia Cell Line, International Journal of Molecular Sciences, 2023, 24(4), 3102;

- Polito R., Very low-calorie ketogenic diet modulates the autonomic nervous system activity through salivary amylase in obese population subjects, International Journal of Environmental Research and Public Health, 2021, 18(16), 8475.

We have added description of the ketogenic diet to the Introduction.

Results

The results are displayed well. If possible, the authors could increase the space, within the same figure, between one graph and another. We increased the horizontal and vertical spacing between graphs for Figure 2.

Paragraph 2.1 should be moved; perhaps it would be more appropriate to include it in the methods section. Our reason for including a description of study design at the beginning of the Results section was because IJMS has the Methods at the end of the manuscript as opposed to in between the Introduction and Results. We thought it was important to put the study into context before jumping into the results as most readers will read from beginning to end and not jump to the Methods.

Discussion

The discussions are well structured; furthermore, the authors also report the limitations of the study and the conclusions. Thank you.

Material and Method

In this section I would move paragraph 2.1 (Study design) that the authors inserted in the results paragraph. Please see explanation above.

Was a test used to evaluate the distribution of the data before carrying out the ANOVA test? Was the data parametric or non-parametric? Please specify this in section 4.5. The following statement has been added to section 4.5: “The vast majority of the data passed the D’Agostino & Pearson test for normal Gaussian distribution and all data were analyzed similarly with parametric ANOVA tests.

Reviewer 3

The manuscript entitled “Ketogenic Diet Affects Sleep Architecture in C57BL/6J Wild 2

Type and Fragile X Mice” by Pamela R. Westmark et al. explored the gene Fmr1 and high fat, low carbohydrates ketogenic diet on sleep acchitecture by using animal models. Although there are no difference between wilf-type and Fmr1ko mice on normal chow diet, but they exhibited difference on ketogenic diets, indicating Fmr1 is partially responsible for the ketogenic diet induced sleep architecture. The study is comprehensive, which should be considered as publishable on the journal. Thank you.

Reviewer 2 Report

The study is very interesting and deals with a very current topic.This study compares sleep states acquired by sleep EEG to published rest-activity cycles attained by actigraphy. The authors find similar sleep-wake patterns in wild type (WT) and Fmr1KO mice maintained on a purified ingredient, casein protein-based diet. KD disrupts sleep-wake patterns in both strains and reveals some genotype-specific differences. Although the manuscript is of interest to the scientific community, I have some suggestions for the authors.

Introduction.

The introduction is a little confusing. The authors move from one topic to another in an unclear manner. For example, in the central part of this paragraph (lines 51-54), the authors seem to want to describe the purpose of the study; subsequently (lines 77-78) the authors report the purpose of the study again. I believe that authors should clearly specify the purpose of the study at the end of the introductory paragraph. Furthermore, in this section, it is not necessary to summarize the results as done in lines 78-81.

I also think the authors could explain the ketogenic diet better; for example they can take inspiration from the following recent publications:

- Polito R., The Ketogenic Diet and Neuroinflammation: The Action of Beta-Hydroxybutyrate in a Microglia Cell Line, International Journal of Molecular Sciences, 2023, 24(4), 3102;

- Polito R., Very low-calorie ketogenic diet modulates the autonomic nervous system activity through salivary amylase in obese population subjects, International Journal of Environmental Research and Public Health, 2021, 18(16), 8475.

Results

The results are displayed well. If possible, the authors could increase the space, within the same figure, between one graph and another.

Paragraph 2.1 should be moved; perhaps it would be more appropriate to include it in the methods section.

Discussion

The discussions are well structured; furthermore, the authors also report the limitations of the study and the conclusions.

Material and Method

In this section I would move paragraph 2.1 (Study design) that the authors inserted in the results paragraph.

Was a test used to evaluate the distribution of the data before carrying out the ANOVA test? Was the data parametric or non-parametric? Please specify this in section 4.5

Author Response

(The authors gave the same response as above.)

Reviewer 3 Report

The manuscript entitled “Ketogenic Diet Affects Sleep Architecture in C57BL/6J Wild 2

Type and Fragile X Mice” by Pamela R. Westmark et al. explored the gene Fmr1 and high fat, low carbohydrates ketogenic diet on sleep acchitecture by using animal models. Although there are no difference between wilf-type and Fmr1ko mice on normal chow diet, but they exhibited difference on ketogenic diets, indicating Fmr1 is partially responsible for the ketogenic diet induced sleep architecture. The study is comprehensive, which should be considered as publishable on the journal.

Author Response

(The authors gave the same response as above.)
